# Neurobiology of Maternal Behavior in Nonhuman Mammals: Acceptance, Recognition, Motivation, and Rejection

**DOI:** 10.3390/ani12243589

**Published:** 2022-12-19

**Authors:** Genaro A. Coria-Avila, Deissy Herrera-Covarrubias, Luis I. García, Rebeca Toledo, María Elena Hernández, Pedro Paredes-Ramos, Aleph A. Corona-Morales, Jorge Manzo

**Affiliations:** 1Instituto de Investigaciones Cerebrales, Universidad Veracruzana, Avenida Luis Castelazo s/n Col. Industrial Ánimas, Xalapa 91190, Mexico; 2Facultad de Medicina Veterinaria y Zootecnia, Universidad Veracruzana, Avenida Miguel Ángel de Quevedo s/n, esq. Yañez, Veracruz 91710, Mexico; 3Facultad de Nutrición, Universidad Veracruzana, Avenida Médicos y Odontólogos s/n Col. Unidad del Bosque, Xalapa 91010, Mexico

**Keywords:** preoptic area, parturition, amygdala, oxytocin, dopamine, recognition, brain, motivation, bond

## Abstract

**Simple Summary:**

Maternal behavior involves active and passive responses associated with the willingness to nurse and protect the young. In some species, its expression is very selective toward individuals that are recognized as their own and may be long-lasting, whereas in other species the expression is not as selective or may be short-lasting. Brain processes of acceptance, social recognition, inhibition of rejection/fear, and increase in care motivation mediate its expression. The neurocircuitry of maternal behavior is activated upon exposure to the right natural stimuli, such as those that occur during pregnancy, parturition, and lactation. However, even virgin females and males can respond with maternal behaviors if they develop sensitization to the offspring via cohabitation or cross-sensitization via mating. Herein, we discuss behavioral expression in different species, the natural triggering stimuli, and the putative neurocircuitries of acceptance, social recognition, motivation, and rejection during maternal behavior.

**Abstract:**

Among the different species of mammals, the expression of maternal behavior varies considerably, although the end points of nurturance and protection are the same. Females may display passive or active responses of acceptance, recognition, rejection/fear, or motivation to care for the offspring. Each type of response may indicate different levels of neural activation. Different natural stimuli can trigger the expression of maternal and paternal behavior in both pregnant or virgin females and males, such as hormone priming during pregnancy, vagino-cervical stimulation during parturition, mating, exposure to pups, previous experience, or environmental enrichment. Herein, we discuss how the olfactory pathways and the interconnections of the medial preoptic area (mPOA) with structures such as nucleus accumbens, ventral tegmental area, amygdala, and bed nucleus of stria terminalis mediate maternal behavior. We also discuss how the triggering stimuli activate oxytocin, vasopressin, dopamine, galanin, and opioids in neurocircuitries that mediate acceptance, recognition, maternal motivation, and rejection/fear.

## 1. Introduction

Survival of the mammal offspring depends on the correct expression of maternal behaviors, particularly during the early postnatal period. Newborns must be a powerful source of incentive sensory stimulation to the dam, and in return, they must be capable of responding either actively or passively to such stimuli, expressing acceptance and motivation to invest energy and time and willingness to risk their physical safety. Hence, the capacity to express maternal behavior depends on the sensitivity to respond to the right stimuli under certain physiological, ontogenic, or cognitive conditions. Accordingly, to understand the neurobiology of this behavior, we must consider neural systems involved in acceptance, social recognition, motivation, and fear/rejection. This review aims to provide information on the neurobiology of such processes in nonhuman mammals. We begin by describing the objective measures of active and passive responses in different species. Then, we discuss the natural stimuli that facilitate the expression of maternal behavior and the putative neurocircuitries. Accordingly, scientific articles related to maternal behavior were analyzed. The electronic databases PubMed, GoogleScholar, and SciELO were searched using the following keywords in English: maternal behavior; animals; preoptic area; parturition; amygdala; oxytocin; dopamine; recognition; brain; motivation; bond. The exploration included studies on laboratory and domestic animals.

## 2. Active and Passive Maternal Behaviors

Maternal behavior involves the facilitation of acceptance, recognition, and motivation, along with the inhibition of rejection and fear toward offspring. Acceptance is inferred from behaviors that allow proximity to any newborn, whereas recognition involves selective acceptance of specific individuals. Thus, females may passively accept unfamiliar newborns (i.e., allowing nursing to any young) and recognize/accept only familiar ones. Similarly, rejection may involve active responses (i.e., aggression/infanticide) to discourage contact, whereas fear may be expressed via passive avoidance (i.e., not approaching them). Furthermore, care motivation involves active behaviors that indicate willingness to nurse and protect the young. So, although the expression of maternal behaviors varies considerably among species, the endpoints served are the same. For instance, some precocial species, such as sheep and horses, express very selective maternal behavior toward offspring they accept/recognize as theirs during the very first hours postpartum [1,2]. That kind of maternal selectivity requires strict mechanisms of acceptance/social recognition that occur via imprinting (i.e., associative learning) during a brief period following parturition. Disturbance of recognition between the dam and her offspring during the early imprinting period may result in rejection (perhaps fear), despite all the hormonal input or whelping experience. By contrast, altricial species with massive reproductive strategies, like rats, canids, or pigs, are considered less strict because they may accept alien offspring during extended periods [3,4,5]. In addition, some ungulates [6] and nonhuman primates [7,8] may display extensive maternal repertoires daily for weeks or months, whereas others, such as lagomorphs, will display only a few minutes of nursing once a day [9].

Some behaviors that start before parturition might be considered indirect maternal behaviors. For example, nest-building (e.g., digging, shredding paper, straw carrying, and hair pulling) and isolation from the pack or herd [10,11,12,13]. Other early behaviors, such as those observed in pregnant dogs, including restlessness, reduced appetite, lack of attention, drowsiness, aggression, anxiety, fickleness, capriciousness, irritation, and increase in attention request, may only reflect an imminent parturition [14,15]. Indeed, nest-building and isolation are associated with searching for and selecting the appropriate birthplace [16,17]. It is possible that nest-building is more likely observed in altricial species, whereas isolation from others may occur in precocial ones.

Direct maternal behaviors are observed after parturition, during the first contact with the newborns. Dogs will actively bite and tear the fetal membranes and cut the umbilical cord, which functions to prevent asphyxiation of the pups [18]. The dam actively licks the head and the mouth of the newborn to stimulate respiration and orient the pups toward the mammary gland [13,15]. They also lick the anogenital area to facilitate urination and excretion during the first 2 postnatal weeks [19]. Rats also express anogenital licking, especially toward males during the first 10 postnatal days [20]. Preference to lick males is evoked by attractive odors from preputial compounds, such as dodecyl propionate [21], which depend on the levels of systemic steroids. If female pups are treated with androgens (i.e., testosterone and dihydrotestosterone) on the day of birth, they receive an equivalent amount of active anogenital licking as males [22]. Enhanced anogenital stimulation appears to have positive long-term effects on male reproductive behavior [23]. Dams display other active responses as well, such as retrieving the pups, oral consumption of the placenta, and defense from predators and conspecifics, and passive behaviors such as huddling and crouching to regulate body temperature or allowing nursing. In the beginning, passive behaviors may depend on acceptance, whereas active behaviors may depend on enhanced motivation to care. Other species, such as cows, also express intense active licking for the first hour and will be very protective if someone approaches. Passive acceptance may occur within the first 30–120 min postpartum when the calf stands and searches for the udder [16] (Table 1).

The lack of maternal behaviors represents a serious problem that jeopardizes not only the survival of the offspring but also a very important mechanism of early socialization, cognitive development, and epigenetic changes associated with resilience to stress [24,25]. Good maternal behavior is associated with the so-called stress hyporesponsive period [24,26], which refers to a delay of the timing of glucocorticoid elevation in infants, associated with reduced stress response in adulthood [26]. Inappropriate maternal behaviors may occur in 50% of primiparous dogs, especially following cesarean section [27] or as a result of early separation during the postpartum period [28].

## 3. Natural Stimuli That Facilitate the Expression of Maternal Behavior

### 3.1. Hormones

Gregarious species have a natural predisposition to care for the young. However, the capacity to express appropriate levels of maternal behavior develops gradually with hormonal changes that occur throughout pregnancy. Then, drastic changes during parturition are needed to trigger the expression of behavior. In rats, for example, concentrations of progesterone (P4) gradually start to increase from the very first day of pregnancy, reaching a peak at day 15, and are followed by a drastic reduction during the last 3 days before delivery. By contrast, the levels of estradiol (E2) and prolactin (PRL) stay relatively low at the beginning but increase dramatically during those last 3 days. The reduction of P4 and increase in hormones such as E2, PRL, oxytocin (OT), and corticosteroids, are the main hormonal drastic changes during parturition [29,30,31] and therefore are associated with sensitization of maternal behavior. For example, in the rabbit doe, digging is stimulated by changes in E2 and P4, while straw carrying and hair pulling are under the control of PRL. In the rat, the reduction in P4 and the increase in E2 and PRL levels facilitate active licking, retrieving, and gathering of pups. Pharmacological blockade of estrogen in the medial preoptic area (mPOA) and small interfering RNA silencing of estrogen receptors (ERα) disrupts maternal behavior in mice [32,33], whereas specific activation of ERα-positive mPOA neurons enhances pup retrieval [34,35]. Likewise, the blockade of PRL receptors within the mPOA in mice abolishes pup retrieval [36].

### 3.2. Vagino-Cervical Stimulation

In the ewe, licking, low-pitched bleats, and nursing are also evoked by changes in P4/E2 ratio and by the release of OT triggered by vagino-cervical (VCS) and nipple stimulation [37]. VCS caused by the passing of the young through the pelvic canal must be considered a powerful triggering stimulus to evoke maternal behavior after hormonal sensitization during pregnancy. For instance, artificial VCS (pressure on and stretching of the neck of the cervix provided by hand) in ewes can facilitate maternal acceptance toward an alien lamb up to 27.5 h postpartum [38,39]. This also occurs in other species, such as rats, in which normal expression of maternal behavior depends on the interaction between hormonal priming in the mPOA and VCS evoked by parturition. One study showed that, 24 h before parturition, only a few pregnant females exposed to pups (from a different female) expressed active pup retrieval, but 12 h before parturition, up to 80% of them retrieved pups. In addition, that study explored the maternal behavior of pregnant females implanted bilaterally in the mPOA with the antiestrogen 4-hydroxytamoxifen (OH-TAM). Accordingly, 12 h before parturition, none of the OH-TAM females expressed retrieving behavior, and in the absence of parturitional experience (delivery by cesarean section), maternal behavior was almost absent upon exposure to their own pups. By contrast, those OH-TAM females that were allowed to undergo normal parturition (with natural VCS) expressed normal retrieving behavior upon exposure to their pups [40]. Hence, hormones and VCS play a synergistic role in evoking the whole repertoire of maternal behaviors. Indeed, pseudopregnant female rats and mice that go through all the hormonal changes without parturition express only a few indirect maternal behaviors such as nest-building [41,42]. Similarly, pseudopregnant dogs (i.e., pseudocyesis) can also express some maternal behaviors (e.g., nesting, defense) toward pup-looking puppets [43]. In one case report, a sudden decrease in systemic P4 following ovariohysterectomy during the luteal phase of diestrus was reported as the triggering stimulus for maternal behavior, evoking a parturition-like drastic reduction of P4. Similarly, sudden maternal behavior has also been observed in pregnant rats following hysterectomy [44]. Males artificially exposed to E2 and P4 also expressed paternal behavior [45]. When those males received lesions in the mPOA, their behavior was significantly reduced, indicating the mPOA mediates parental behavior in both males and females.

Hormones and physical stimuli (VCS, nipple stimulation) that occur during parturition and lactation are the best natural stimuli that induce maternal behavior. Upon stimulation, magnocellular neurons in the supraoptic (SON) and paraventricular nuclei (PVN) fire high-frequency bursts of action potentials. Each burst generates a large pulse of OT release into the bloodstream to evoke contractions of the uterus and milk ejection [46]. Likewise, parvocellular neurons release OT toward the central nervous system. As our study shows, OT modulates acceptance, social recognition, learning, memory, emotions, reward, eating, drinking, sleep, wakefulness, nociception, analgesia, and sexual and maternal behaviors [47].

### 3.3. Exposure to Pups

Interestingly, exposure to pups can also result in sensitization of maternal behavior in male and nonpregnant female rats. One week of daily exposure to pups induces both active (e.g., nest-building, retrieving, licking) and passive behaviors (e.g., nursing posture) [48,49,50]. This indicates that gradual exposure sensitizes parental behavior without the need for any hormonal priming. This type of sensitization also occurs when juvenile rats are exposed to infant rats [51], and watching a conspecific perform maternal behavior (i.e., retrieval) activates OT neurons in the observer [52]. Accordingly, the putative neurocircuitry that mediates maternal behavior might be gradually activated and sensitized by daily exposure to pups, but hormones, parturition, and lactation function as triggering stimuli that accelerate its activation.

### 3.4. Mating

Similarly, copulation can also sensitize the neurocircuitry of parental behavior. In male rats [53] and mice [54], copulation blocks infanticide behavior (an expression of rejection) and facilitates active retrieving (an expression of care motivation) in future encounters with pups [55]. More than 90% of male mice will normally commit infanticide if exposed to pups between 1–4 days after mating with any female, indicating that during those immediate days males reject the pups. However, between 80–90% of those males will behave parentally and will not kill the pups if they are exposed to them 12–50 days after copulating to ejaculation. The actual mechanisms for sex-induced parental behavior in males are unknown but appear to require changes in the mPOA. This area is sensitive to mounts, intromissions, and ejaculations [56,57], such that consecutive copulatory series increase the number of firing neurons in the mPOA [58] and lesions impair consummatory sexual behavior [59]. Those changes may modify plasticity within the mPOA to facilitate parental behavior.

### 3.5. Maternal Experience

Former maternal experience also improves the expression of maternal responses. For example, multiparous dogs express more time of body contact with pups and constant maternal care during the 21-day postpartum period, whereas primiparous females show a gradual increase in licking, nursing, and contact with the puppies from day 1 to 21 [60]. Multiparous cows also express more maternal defense than primiparous [61] and isolate less from the herd [62], probably related to less intense fear, considering that multiparous female rats are less anxious in open field tests, compared to primiparous females [63]. In sheep, former maternal experience is associated with increased suckling, following, grooming, and low-pitched bleating and decreased aggressive behavior [64].

### 3.6. Environmental Enrichment

Environmental enrichment (EE) also improves maternal behavior. In one study with rats, EE condition consisted of housing seven females per cage; the EE cage (120 × 100 × 70 cm) was designed with four floors with lid ramps and contained plastic balls, tubes, and bedding material. The interactive objects and location of food were rearranged every three days to increase novelty and complexity, which resulted in a highly stimulating sensory and social environment with other females. Following parturition, EE females expressed less anxiety and displayed more licking, grooming, and crouching over pups during the first postpartum week as compared to females living in standard cages [65]. During that period, EE mothers also showed more aggressiveness to an intruder female. Associated with offspring-directed behaviors, EE females expressed more neural activity in the mPOA, PVN, and medial amygdala (MeA) but less activity in the basolateral amygdala (BLA) than standard-housed females [66]. As we discuss in the following sections, those brain areas are associated with maternal motivation and rejection, respectively. Taken together, the data indicate that the capacity to express maternal behavior develops as a consequence of hormonal priming but is triggered by stimuli such as parturition and lactation. In addition, cohabitation, copulation, former experience, and environmental enrichment facilitate its expression (Table 2).

## 4. Neural Pathways Underlying Maternal Behavior

### 4.1. Areas Involved in Acceptance and Social Recognition

Maternal behavior requires at least three neural processes to mediate: (1) increase in acceptance/social recognition, (2) decrease in rejection/fear, and (3) increase in care motivation [67]. Acceptance/recognition occurs when a sensitized brain is exposed to the right stimulus. For example, during the postpartum period, bitches will accept any pup scented with amniotic fluid. Washing the pups immediately after birth results in a lack of acceptance. However, if pups are bathed in amniotic fluid, the process of acceptance is restored, even hours later [28]. Similarly, washing a newborn lamb impairs acceptance/recognition behaviors in sheep (low bleats, acceptance at the udder, nursing, and licking time) and increases rejection/fear behaviors (high-pitched bleats, rejection at the udder, and aggressive behavior). Preventing contact with the mother during the first 4 h postpartum worsens rejection. Interestingly, washing an alien lamb also improves its acceptance [68]. Indeed, both acceptance and rejection behaviors depend on the olfactory system because anosmic ewes (treated with intranasal zinc sulfate) fail to express any acceptance or rejection behavior [69]. However, if tested many hours after birth, anosmic sheep will be capable of accepting/recognizing their lamb through visual or auditory cues that have been learned during the first postpartum hours [70].

Olfactory social recognition is perhaps the most relevant sensory system in many mammals and depends on both main and accessory olfactory pathways. Nonvolatile odor molecules are detected by the accessory pathway via the vomeronasal organ (VNO), located in the soft tissue of the nasal septum. The VNO projects to the accessory olfactory bulb (AOB), which innervates both the bed nucleus of the stria terminalis (BNST) and medial amygdala (MeA). BNST and MeA project to the mPOA, which in turn projects to the lateral preoptic area (LPOA) and substantia innominata (SI). LPOA and SI also send efferents to the ventral tegmental area (VTA), which sends dopaminergic projections to the nucleus accumbens (NAcc) [71,72]. With regard to volatile odor molecules (those that have a low boiling point and evaporate easily at room temperature), they are detected in the roof of the nasal cavity by the main olfactory epithelium (MOE), where olfactory sensory neurons are located. These neurons project to the main olfactory bulb (MOB), and MOB has efferents to the piriform cortex (PirCtx), and from there to the NAcc and mPOA [71]. Thus, both volatile odors and nonvolatile odors not only access the social recognition system but also the mesolimbic pathway of motivation. Within these pathways, social recognition depends on the nonapeptides arginine vasopressin (AVP) and oxytocin (OT). Mice with depletion of AVP V1a receptor (V1aRKO) exhibit profound social recognition impairment. They can remember the odors of food but fail to remember the odors of individuals they just spent time with [73,74]. Likewise, OT knockout mice (OTKO) express impaired discrimination to odors of conspecifics [75]. Following a social encounter, OTKO mice express less activity in MeA, BNST, and mPOA, but an infusion of OT in the MeA is sufficient to restore social recognition [74,76,77]. Interestingly, during the postpartum period, OT receptor (OTR) binding increases not only in the MeA but also in the BNST, mPOA [78], and dopaminergic regions of the brain stem [74,79,80,81,82,83]. The increase in OTR is mainly the result of exposure to E2 [84]. Thus, OT and OTR start to play a gradual role in the acceptance and olfactory social recognition via MeA starting two days before parturition when E2 increases. Experiments in sheep have shown that infusion of local anesthesia in the MeA or cortical amygdala (CoA) impairs recognition of the lamb during the early postpartum period, but this was not found with infusions in the BLA region [85]. As we describe later, BLA mediates rejection/fear responses, whereas MeA and CoA mediate olfactory acceptance/recognition, mainly influenced by OT, AVP, and dopamine (DA) (Figure 1).

In dogs, salivary OT (sOT) increases gradually during the postpartum period [86]. High levels of sOT are negatively correlated with the frequency of sniffing behavior toward the pups, perhaps because high sOT facilitates olfactory recognition, and recognized pups require less sniffing. Given that sniffing and time spent out of the whelping box have been positively correlated, low sOT is also a predictor of time away from the pups. Therefore, treatment with OT should improve maternal recognition and performance. In fact, rats that undergo cesarean section lack a naturally powerful stimulus to release OT, which results in reduced maternal behavior. However, intranasal OT in those rats restores pup retrieval and anogenital licking [87]. Similarly, intranasal OT facilitates paternal motivation, as observed in male marmosets that express shorter latency to respond with an approach to infant stimuli [88]. Dogs treated with intranasal OT turn more social toward humans [89] and more playful with other dogs [90]. In female rats, OT improves depression-like behaviors [91] and induces conditioned place preference (CPP) in the presence of a conspecific (social-CPP). Social-CPP reflects emotional memories of a place where cohabitation occurred, and it has been reported in rats that receive MDMA “Ecstasy” but not in those that receive AVP. Accordingly, both OT and AVP regulate olfactory recognition, but only OT augments the rewarding effects of social interaction [92].

In dogs, polymorphisms of the OTR gene have been associated with variations in the maternal behavior and with higher levels of sOT [93]. However, to date, no one has shown the potential good effects of intranasal OT in dogs with poor maternal behavior. Certainly, systemic injections of OT at birth are commonly indicated to facilitate uterine contractions, but only 1% of such systemic OT crosses the blood–brain barrier (BBB) [94]. Nevertheless, it has been demonstrated that fetal voles from mothers injected with OT at birth expressed increased methylation of OTR in the brain. In adulthood, OT-exposed voles are more gregarious, with increased alloparental caregiving toward pups and increased close social contact with other adults [95]. Thus, OT treatment at birth (even if injected) may facilitate both immediate and future social encounters.

### 4.2. Areas Mainly Involved in Increasing Motivation

The NAcc and mPOA are the main areas involved in maternal motivation [96]. In rats, broad lesions in the mPOA disrupted pup retrieval and reduced Fos-immunoreactivity in the NAcc evoked by exposure to pups [97]. However, specific lesions in the ventromedial region of the preoptic area (vmPOA) disrupt nest-building in mice but do not affect pup retrieval, whereas specific lesions in the central part of the mPOA (cmPOA) disrupt all maternal behaviors [98]. Lesions in the NAcc also decrease pup retrieval and bar pressing to gain access to pups [99], whereas lesions in the VTA (where NAcc DA afferents originate) disrupt the frequency of approaches and interaction with pups [100,101].

Between mPOA and NAcc, there is a synergistic role OT and DA. One study showed that female rats that display high levels of active maternal behavior (i.e., licking/grooming) express more OT-positive neurons in the mPOA and PVN, which in turn, project to the VTA. Thus, OT in the VTA facilitates DA release into the NAcc, which also enhances motivation [102]. The mPOA itself is a large and complex region formed by many subnuclei, such as the medial preoptic nucleus (MPN), median preoptic nucleus (MnPO), posterodorsal preoptic nucleus (PD), and ventrolateral preoptic nucleus (VLPO) in mice. Within all these nuclei, there is a vast diversity of neuron populations based on gene expression for transporters or neurotransmitters that are not found in the lateral preoptic area (LPOA). For instance, at least fifteen different markers of cell populations have been reported in the mice mPOA, including glutamic acid decarboxylase (GAD67), galanin, calbindin, a novel marker of sexually dimorphic nuclei (Moxd1), cocaine- and amphetamine-regulated transcript (CART), vesicular glutamate transporter (VGLUT2), vesicular GABA transporter (Vgat), proenkephalin (Penk), brain-derived neurotrophic factor (BDNF), leptin, cholecystokinin (CCK), neurotensin, neuropeptide tachykinin 2 (Tac2), prodynorphin (Pdyn), thyrotropin-releasing hormone (TRH), and oxytocin (reviewed in [103]). Many of these markers are known for controlling mechanisms of body temperature, wake–sleep cycle, or sexual behavior and appear not to have any direct role in maternal behavior. However, galaninergic neurons within the lateral part of the mouse mPOA (lmPOA) and MPN project toward MeA, VTA, PVN, and periaqueductal gray (PAG) and play a major role in motor coordination, motivation, and social recognition during maternal behavior by integrating inputs from many areas in the brain [104]. More than 70% of galanin-positive neurons in the mPOA also express alpha estrogen receptors (ERα) and androgen receptors [105] and therefore may become more active during the final phase of pregnancy. Optogenetic activation of mPOA galanin-positive neurons projecting to the PAG enhances pup grooming, whereas inhibition of those neurons impairs grooming. Likewise, activation of galanin neurons that project to VTA promotes interaction with pups but not pup retrieval, and their inhibition suppresses interaction [104]. The mPOA also receives input from hypocretin-1-containing neurons (HCRT-1, also known as orexin A) from the postero-lateral hypothalamus. Activation of HCRT-1 metabotropic receptors within the mPOA promotes active maternal behaviors such as licking and retrieving but also passive responses such as nursing [106]. The mPOA also receives inhibitory inputs via agouti-related neuropeptide (AGRP) neurons from the arcuate nucleus, which respond in conditions of caloric needs and normally produce hunger. Activation of AGRP inhibitory neurons or its projections to mPOA results in less maternal nest-building, without affecting pup retrieval, partly recapitulating suppression of maternal behaviors during food restriction [107]. Thus, retrieving pups and nest-building are associated with different neuronal activity patterns in the mPOA, such that neurons that are activated during pup retrieval tend to be inhibited during nest-building [34]. Some AGRP neurons that project directly to mPOA express Vgat. Optogenetic stimulation of Vgat neurons in the mPOA elicits both pup retrieval and nest-building, whereas inhibition of Vgat neurons results in decreased nest-building. Thus, GABA activation in mPOA can inhibit inhibitory AGRP neurons and facilitate all maternal behaviors. However, if stimulation is exclusively directed toward the subpopulation of Vgat neurons that express ERα, then only pup-retrieval is elicited, and not nest-building [107] (Table 3). In addition, ERα-positive neurons in the mouse mPOA project inhibitory efferents to nondopaminergic neurons in the VTA that promote maternal pup retrieval through disinhibition of dopaminergic neurons [34]. Accordingly, galanin-positive, ERα-positive, and Vgat-positive neurons may be involved in maternal motivation (pup retrieval) via mPOA and VTA [103] (Figure 2).

Like the mPOA, the NAcc also facilitates motivation toward recognized/accepted individuals. The NAcc is also important for the consolidation of social bonds. For example, neurons of the NAcc in monogamous voles express more OT receptors than in the polygamous voles [108]. Infusion of an OT antagonist into the NAcc of monogamous voles disrupts pair bonds facilitated by DA agonists, whereas OT antagonists disrupt bonds facilitated by DA agonists [109]. Pair bonds that develop after sex are disrupted by infusions of DA antagonists (i.e., haloperidol) in the NAcc, and by contrast, low doses of DA agonists (i.e., apomorphine) facilitate their development even in absence of sex [110,111]. Thus, having a working DA system is necessary, but not sufficient (OT is needed) to induce selective social bonds. During exposure to pups, maternal rats release more DA in the NAcc [112], as well as during licking and grooming of the pups [113]. Likewise, infusions of general DA antagonists such as flupenthixol into the NAcc of lactating rats impair maternal behavior (retrieval and licking) [114]. Specific injections of a D1-type antagonist (SCH23390) in the NAcc impairs pup retrieval, whereas a D2-type antagonist (eticlopride) does not impair such behavior [115]. Thus, maternal behavior is mainly facilitated by D1-type receptors and OT.

### 4.3. Areas Mainly Involved in Reducing Rejection/Fear

In addition to being motivated, individuals must reduce their rejection and fear of pups during the first postpartum encounters. This represents a dichotomy in excitatory/inhibitory neurocircuitries and appears to depend on the positive interaction of mPOA/BNST and a negative mechanism mediated by the anterior hypothalamic nucleus (AHN)/ventromedial nucleus (VMN)/PAG, amygdala, and some cortical areas [116]. Thus, activation of mPOA appears to facilitate maternal behavior upon its release from the inhibitory effects of the amygdala [117]. For instance, lesions of the vomeronasal inputs (amygdala) to the mPOA facilitate maternal behavior even in nulliparous female rats [118,119,120], mainly due to BLA projections that mediate fear responses [121,122]. BLA increases its expression of OTR during the postpartum period [108], and it can be inhibited by endogenous opioids as well [123]. Therefore, the elevation of OT and endogenous opioids during parturition (i.e., during VCS) may help reduce fear/rejection toward pups [124]. Opioids also regulate the release of both OT [125] and DA into the prefrontal cortex (PFC) [126]. PFC plays a key role in attention, memory, and negative emotions [127]. When activity in PFC is reduced, sheep express less aggressive behavior toward alien lambs without affecting the social recognition of their own lambs [128]. PFC also expresses more ERα and OTR in lactating rats, which appear to be associated with decreased anxiety [129].

The central region of mPOA and MPN also mediate the inhibition of some rejection responses. These regions express galanin, and direct optogenetic stimulation of galanin-positive neurons in male mice suppresses pup-directed aggression [130]. Within the vmPOA, some neurons mediate sickness symptoms [131]. However, it is unknown whether those neurons may induce rejection or fear toward pups.

**Figure 2 animals-12-03589-f002:**
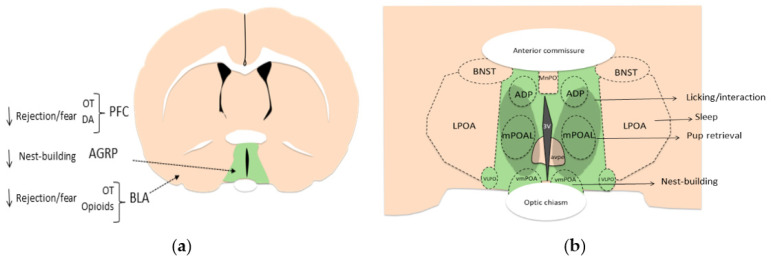
(**a**) Coronal drawing of a rat brain (modified from [132]). The green region represents the medial preoptic area (mPOA) as the main generator of maternal motivation via oxytocin (OT) and dopamine (DA) activity. Agouti-related peptide (AGRP) facilitates rejection in mPOA, but OT, DA, and opioids reduce rejection via other regions such as prefrontal cortex (PFC, not shown) and basolateral amygdala (BLA). (**b**) Amplification of the mPOA and subregions in the rat brain. Darker green spot represents putative galanin/ER+ neurons that mediate the expression of licking and pup retrieval in the lateral (mPOAL) and dorsal (ADP) parts of mPOA, where OT neurons are also found (modified from [103]). Nest-building depends on the activity of the ventral portion of the mPOA (vmPOA). Also shown are the bed nucleus of stria terminalis (BNST), median preoptic nucleus (MnPO), ventrolateral nucleus (vlPO), anteroventral periventricular nucleus (avpv), and lateral preoptic area (LPOA) that mediates sleep. Different mPOA subregions mediate diverse maternal behaviors interacting with mesolimbic regions.

**Table 3 animals-12-03589-t003:** Brain areas and neurochemicals associated with maternal behaviors. Medial amygdala (MeA), bed nucleus of stria terminalis (BNST), cortical amygdala (CoA), medial preoptic area (mPOA), ventral tegmental area (VTA), basolateral amygdala (BLA), nucleus accumbens (NAc), prefrontal cortex (PFC), oxytocin (OT), estrogen (E2), prolactin (PRL), dopamine (DA), vesicular GABA transporter (Vgat), hypocretin-1 neurons (HCRT-1), agouti related protein (AGRP).

Brain Area	Neurochemical	Effect	Representative References
MeA BNST CoA	OT	Acceptance, social recognition	[74,79,83]
mPOA	E2 PRL OT DA Galanin Vgat HCRT-1 AGRP	Pup retrieval Pup retrieval, nest-building Acceptance, licking/grooming Motivation, licking, grooming Motor coordination, motivation, recognition, grooming, inhibition of aggression Nest-building Licking/retrieval Inhibition of nest-building	[32,33,40] [36] [102,103] [103,104,130] [130] [106] [107]
VTA	OT	Approach, interaction	[74,79,86]
BLA	OT opioids	Inhibit rejection/fear	[85,108,121,124]
NAc	DA OT	Motivation, approach, interaction, pup retrieval	[99,100,101,112]
PFC	OT DA	Inhibit rejection/fear	[125,126]

## 5. Conclusions

The capacity to express maternal behavior develops gradually as a consequence of hormonal priming during pregnancy. Around parturition time, sudden changes in hormones and physical stimuli trigger the expression of acceptance, recognition, and motivation at a time that reduces rejection and fear. The neurocircuitries of maternal behavior can be also sensitized by cohabitation, sex, and experience. Olfactory recognition depends mainly on the activity of OT in the MeA. Motivation is mainly mediated by OT and DA in NAcc and subregions of the mPOA, in which ventral parts mediate nest-building, and central and dorsal parts mediate licking retrieval and interaction with pups. Finally, BLA, PFC, and other hypothalamic regions (i.e., arcuate nucleus) mediate rejection and fear responses. These areas mediate discrimination of stimuli that might have an incentive value per se (i.e., amniotic fluid) or stimuli related to past experiences (i.e., memories of sexual reward, maternal reward, cohabitation, etc.).

## Figures and Tables

**Figure 1 animals-12-03589-f001:**
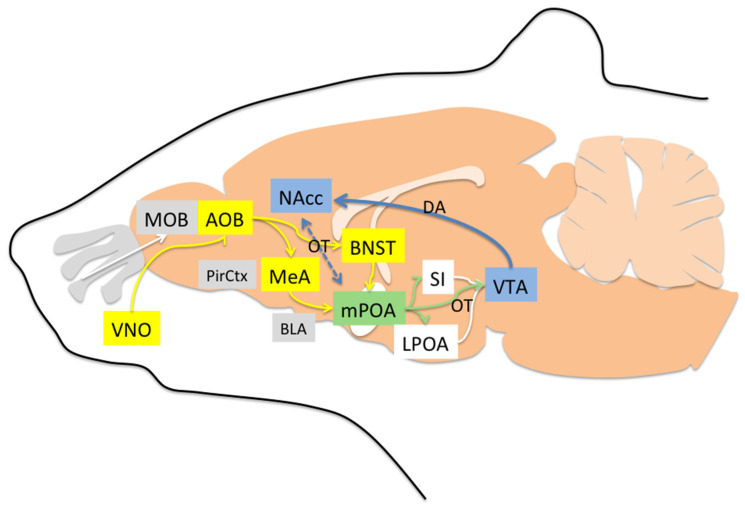
Sagittal drawing of brain olfactory pathways involved in acceptance, offspring social recognition, care motivation, and inhibition of rejection. Nonvolatile odor molecules are processed by the accessory pathway (yellow) via the vomeronasal organ (VNO), accessory olfactory bulb (AOB), bed nucleus of the stria terminalis (BNST), medial amygdala (MeA), and medial preoptic area (mPOA, green). The mPOA projects to the lateral preoptic area (LPOA), substantia innominata (SI), and ventral tegmental area (VTA, blue), which sends dopaminergic projections to the nucleus accumbens (NAcc, blue). Volatile odor molecules are processed by the main (gray) olfactory epithelium (MOE), main olfactory bulb (MOB), and piriform cortex (PirCtx), and from there to the NAcc and mPOA (which interact via oxytocin “OT” and dopamine “DA”). The basolateral (BLA) amygdala and prefrontal cortex (not shown) participate in rejection and fear.

**Table 1 animals-12-03589-t001:** Examples of active and passive maternal behaviors that are associated with processes of acceptance, recognition, motivation, and rejection/fear.

Responses	Acceptance	Recognition	Motivation	Rejection/Fear
Passive	Allow proximity	Selective proximity	Staying in nest	Avoidance
	Nursing	Selective nursing	Crouching over	
Active		Directed calls	Licking	Aggression
		Directed attention	Retrieving	Infanticide
			Placenta consumption	
			Defense	
			Nest-building	

**Table 2 animals-12-03589-t002:** Natural stimuli that trigger maternal behavior. Estrogen (E2), progesterone (P4), prolactin (PRL), oxytocin (OT), vaginocervical stimulation (VCS). ⇑ = increase, ⇓ = decrease.

Stimuli	Effect	Behavioral Response	Representative References
Hormones E2 P4 PRL OT	⇑ ⇑ ⇑ ⇑	Pup retrieval Retrieving Nest-building Nursing	[34,35] [45] [36] [47]
Parturition VCS	⇑	Nursing Retrieving	[38,39] [40]
Mating	⇓	Infanticide	[54,55]
Exposure to pups	⇑	Nest-building, retrieving, licking, crouching posture	[48,49,50]
Experience Multiparous	⇑ ⇑ ⇑ ⇑ ⇓	Licking, nursing, contact Defense Suckling Following, grooming Rejection	[60] [61] [64]
Environmental enrichment	⇓ ⇑	Anxiety Licking, grooming, crouching posture, defense	[65] [66]

## Data Availability

Not applicable.

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
