# Peer review of "Neurobiology of Maternal Behavior in Nonhuman Mammals: Acceptance, Recognition, Motivation, and Rejection"

_animals, 2022, doi:10.3390/ani12243589_

Round 1

Reviewer 1 Report

This manuscript explains how the olfactory pathways, the interconnections of the medial preoptic area with structures such as nucleus accumbens, ventral tegmental area, amygdala, and bed nucleus of stria terminalis mediate maternal behavior in non-human mammals. The manuscript also discusses how the triggering stimuli activate oxytocin, vasopressin, dopamine, galanin and opioids in neurocircuitries that mediate acceptance, recognition, maternal motivation and rejection. I consider that this manuscript is written well. Possible neural pathways and important brain nuclei are summarized well in Figures 1 and 2. It will be nice to prepare tables that indicate the neuropeptides, neurotransmitters and their receptors acting in these pathways and brain nuclei. 

Minor comment

Line 138: Add “stimulation” before (VCS).

Author Response

Thank you for your comments. We have now included a table with the requested information on neurochemicals that participate in maternal behavior (table 3).

Regarding line 138, we found the following sentence: “In the ewe, licking, low-pitched bleats, and nursing are also evoked by changes in P4/E2 ratio and by the release of OT triggered by vagino-cervical (VCS) and nipple stimulation”. We consider that the word “stimulation” may not be required because it is already in that sentence.

Reviewer 2 Report

The Review summaries the current information about the neurobiological background of maternal behavior and provides a useful overview of the field

The first 2 chapters contain are a lot of rough general overviews and the connection to neurobiological pathways is not that obvious. Here, one could emphasize the relationship to neurobiology more strongly. Also, a large  group of animals, all non-human mammals, are covered. Especially in the first two chapters it is not always clear, if the stated things are true for the whole group, or if it is only shown for a particular species.

Line 78 and 79: It would be good to separate which animals show clear nest building behavior. There might be a difference between precocial and altricial animals?

Line 82-83. Separation of the pack is already mentioned in line 79. This might be combined.

3.6 Environmental enrichment: This chapter is rather short. There is a lot of information available regarding environmental enrichment and it’s influence on the brain. I am not an expert in neurobiology, but the authors should make sure that nothing important is overlooked here. There are also various forms of enrichment which seems to cause different effects. I think the chapter could benefit from specifying the form of enrichment more precisely here.

Figure 2 is messed up.

Effect of stress seems a bit underrepresented.

Author Response

Response: Thank you for your comments.

We have tried to categorize the topic in sections.

We keep in mind that many readers of the article may be more familiar with domestic species, and not necessarily with laboratory models of maternal behavior. We find the comparison very useful and prefer to keep it that way.

Indeed, your query about nest-building behavior and its relation to being altricial or precocial is interesting.

We have added a paragraph about it (section 2, lines 90-).

We have corrected the lines of separation behavior and have added information to the environmental enrichment paragraph.

Figure 2 looks good in the manuscript we submitted. We do not know what happened.

Reviewer 3 Report

This article is generally interesting and well written. My minor comments aim to increase the scientific soundness and clarity of it.

Line 53 – A brief description of the methodology of the literature analysis is missing.

Line 335 – markers of what?

Page 354 – the authors implement very awkward way to describe the character the nature of neurons. For example, they use terms like “hypocreatin-1 neurons” or “galanin neurons”. In fact, there are “galaninergic neurons” or “hypocretin-1-containing neurons”. The same is true in relation to other biologically active substances.

Line 358 – the abbreviation for arcuate nucleus is ARC

Figure 2 – please correct this figure. It is illegible in its present form.

Line 408, 412 – PAG and BLA were already abbreviated in line 345 and line 395.

Author Response

Response: Thank you for your comments.

We have added a paragraph on the methodology of the literature analysis.

We have clarified we refer to markers of cellular populations.

We have corrected the nomenclature of types of neurons.

We do not abbreviate arcuate.

Figure 2 looks ok in our submmitted manuscript. We do not know what happened.

We corrected the abbreviations of PAG and BLA.

Round 2

Reviewer 1 Report

I consider that the authors have adequately revised the manuscript.